# Localization of Vestibular Cortex Using Electrical Cortical Stimulation: A Systematic Literature Review

**DOI:** 10.3390/brainsci14010075

**Published:** 2024-01-11

**Authors:** Christina K. Arvaniti, Alexandros G. Brotis, Thanasis Paschalis, Eftychia Z. Kapsalaki, Kostas N. Fountas

**Affiliations:** 1Department of Neurosurgery, University Hospital of Larissa, Faculty of Medicine, School of Health Sciences, University of Thessaly, 41110 Larissa, Greece; arvanitixristina@hotmail.com (C.K.A.); alexgbrodis@yahoo.com (A.G.B.); 2Department of Neuro-Oncology, Cambridge University Hospital, Cambridge CB4 1GN, UK; thanospaschalis@gmail.com; 3Department of Diagnostic Radiology, University Hospital of Larissa, Faculty of Medicine, School of Health Sciences, University of Thessaly, 41100 Larisa, Greece; rad.kapsal@gmail.com; 4Advanced Diagnostic Institute Euromedica-Encephalos, 15233 Athens, Greece; 5Faculty of Medicine, University of Thessaly, Biopolis, 41110 Larissa, Greece

**Keywords:** vestibular, vertiginous, cortex, current intensity, stimulation, vertigo

## Abstract

The vestibular system plays a fundamental role in body orientation, posture control, and spatial and body motion perception, as well as in gaze and eye movements. We aimed to review the current knowledge regarding the location of the cortical and subcortical areas, implicated in the processing of vestibular stimuli. The search was performed in PubMed and Scopus. We focused on studies reporting on vestibular manifestations after electrical cortical stimulation. A total of 16 studies were finally included. Two main types of vestibular responses were elicited, including vertigo and perception of body movement. The latter could be either rotatory or translational. Electrical stimulation of the temporal structures elicited mainly vertigo, while stimulation of the parietal lobe was associated with perceptions of body movement. Stimulation of the occipital lobe produced vertigo with visual manifestations. There was evidence that the vestibular responses became more robust with increasing current intensity. Low-frequency stimulation proved to be more effective than high-frequency in eliciting vestibular responses. Numerous non-vestibular responses were recorded after stimulation of the vestibular cortex, including somatosensory, viscero-sensory, and emotional manifestations. Newer imaging modalities such as functional MRI (fMRI), Positron Emission Tomography (PET), SPECT, and near infra-red spectroscopy (NIRS) can provide useful information regarding localization of the vestibular cortex.

## 1. Introduction

The importance of the vestibular system in various human body functions is undoubtedly paramount [1]. It plays a fundamental role in body orientation (horizontal and vertical), posture control, spatial perception, body motion perception, conception of object location, gaze, and eye movements [2,3]. Configuration of all the anatomical components of the human vestibular system, as well as its interconnections, is of paramount importance not only for understanding its functions but also for explaining the pathophysiology of various clinical disorders and symptoms affecting it [2,3]. It has been widely accepted that the vestibular pathways connect the VIII cranial nerve to the ocular motor nuclei and the supranuclear integration center of the rostral midbrain via the medial longitudinal fasciculus [2,4,5]. It is also known that axons from the VIII cranial nerve rely on thalamic nuclei, and then their thalamic projections terminate in various, vaguely-defined, cortical areas [5].

Knowledge of the exact anatomic location of the cortical areas involved in the processing of the vestibular stimuli in humans remains controversial, while several investigators have questioned the existence of such a vestibular cortex [1,6,7,8,9,10,11,12]. Evidence from studies on humans is limited [12]. It usually arises from diverse clinical studies in epilepsy patients after electrical cortical stimulation, and from activation studies using functional neuroimaging and/or magnetoencephalography [12,13,14]. Among these studies, there is significant heterogeneity in the studied population, their design, and outcome parameter to be recorded, with highly variable accuracy characteristics [12,14]. As a result, a number of cortical areas have been implicated in processing the vestibular stimuli, including the mesial temporal areas and the neocortical temporal, parietal, and occipital lobes [12,14]. To make things more complex, a number of clinical manifestations, such as presyncope and epilepsy, may mimic vestibular responses [15]. Recently, it has been proposed that the vestibular cortex is more like a cortical network rather than a specific region with unique functional areas [12].

In our current review, our primary aims were to describe the elicited vestibular responses after electrical stimulation of cortical structures (Q1) and identify and summarize any potential anatomic correlations (Q2). Secondary aims were to describe the potential role of the employed stimulation intensity and frequency in eliciting certain vestibular responses (Q3), as well as to identify any non-vestibular responses after the electrical stimulation of the cortical areas associated with vestibular functions (Q4).

## 2. Material and Methods

### 2.1. Study Design

We conducted a literature-based, systematic review focusing on vestibular responses after electrical cortical stimulation. The objectives, methodology, and the inclusion criteria for enrollment in this systematic review were pre-specified in a standardized protocol. Our current manuscript was prepared according to the Standard Preferred Reporting Items for Systematic Reviews and Meta-Analyses (PRISMA) guidelines [16]. Our systematic review was not registered at any database.

### 2.2. Search Strategy

An electronic search of the English-language literature was conducted until November 2023 using the PubMed, Scopus, and Web of Science databases. We used the following terms and their variants: “direct current stimulation” OR “electrodes” AND “Vestibular” OR “vertiginous” AND “cortex” (Table 1). Further articles were also searched via thorough examination of the references cited in the initially identified reports.

### 2.3. Eligibility Criteria

Our search extended to all types of studies, including randomized controlled studies, observational studies, and case series. However, our search was limited to studies in English, while we excluded editorials and reviews. Independently, two reviewers (AB and TP) evaluated the eligibility of studies for inclusion in this review, in a non-blinded, standardized manner. In the case of disagreement, all issues were discussed with the senior author (KF).

### 2.4. Data Extraction Evidence Synthesis

Each study was described by the name of the primary author and the year of publication. We extracted the following data from each article: the country of origin, its design, the study population, the employed stimulation protocol, and the primary outcome. A summary of all findings related to the current review was written after a consensus of all the authors.

### 2.5. Quality Evaluation

Two review authors (TP and AB) stratified the gathered studies according to their design into randomized controlled studies, observational studies, and case series, with a descending order regarding the quality of evidence.

## 3. Results

### 3.1. Literature Search

A total of 363 studies were found after the electronic literature search, while 3 additional studies were identified through the reference lists of the retrieved articles (Figure 1). Two hundred and fourteen articles remained after removing all duplicate studies. After reading the title and the abstract, 124 studies were found to be irrelevant and were discarded. From the remaining 90 studies, 77 studies were excluded after reading the full-text manuscript. The three additional studies identified through citation searching were included in the review. Finally, 16 studies reporting on the electrical stimulation of cortical areas associated with vestibular functions were selected (Table 2).

### 3.2. Quality of the Evidence

There was no randomized controlled study among our eligible studies. All included studies were retrospective in nature. There was one cohort study [17], twelve case-series [2,12,18,19,20,21,22,23,24,25,26,27], and three case reports [28,29,30]. As a result, the gathered evidence was based on low and very-low quality of evidence [31]. It has to be pointed out that there was a potential population overlap in three studies by Mazzola et al. [17,23,24]. We decided to include all of them in our study, since they differed in their outcome analysis.

**Table 2 brainsci-14-00075-t002:** **Summary of the basic characteristics of the included studies**.

**Penfield (1957, Canada)** [2]	
**Methods**	Retrospective case series
Participants	108 patients undergoing exploratory craniotomy
Interventions	Gentle electrical stimulation of the cerebral cortex
Outcome	Vestibular sensation
Notes	Stimulation of the transverse gyrus, the parietal lobe, and occipital cortex elicited dizziness, sensation of rotation or body displacement, and conjugate deviation of the eyes with dizziness, respectively
**Richer (1993, Canada)** [18]	
Methods	Retrospective case series
Participants	40 patients undergoing preoperative evaluation for DRE
Interventions	Bipolar stimulation of the somatosensory cortex (rolandic and parietal regions)
Outcome	Somatosensory responses
Notes	3 out of 12 stimulation sites in medial parietal regions elicited perceptions of altered body position
**Fish (1994, Canada)** [19]	
Methods	Retrospective case series
Participants	75 patients undergoing preoperative evaluation for DRE
Interventions	Bipolar stimulation of the temporal and frontal lobes
Outcome	Responses without after-discharge spreading beyond the site of stimulation
Notes	Vague cephalic sensations or dizziness occurred in seven patients, in one with hippocampal stimulation, and in six with stimulation of the amygdala. Five out of seven patients reported similar findings during spontaneous seizures
**Salanova (1995, Canada)** [20]	
Methods	Retrospective case series
Participants	80 patients with parietal lobe epilepsy
Interventions	Parietal lobe resections
Outcome	Clinical manifestations and outcomes
Notes	During intra-operative monopolar cortical stimulation of the parietal lobe, one patient reported a sensation of ‘rolling’ off the table. Two patients developed a disturbance of body image; one felt a twisting sensation in the contralateral extremity, and another stated that ‘I just swayed’ following stimulation of the non-dominant inferior parietal lobule
**Blanke (2000, Switzerland)** [28]	
Methods	Retrospective case report
Participants	1 patient undergoing preoperative evaluation for DRE
Interventions	Electrical cortical stimulation using subdural grid electrodes covering the covering parts of the left frontal, parietal, and temporal lobe
Outcome	Vestibular responses
Notes	Vestibular responses were evoked after stimulation of the inferior parietal lobule at the anterior part of the intraparietal sulcus
**Kahane (2003, France)** [12]	
Methods	Retrospective case series
Participants	260 patients undergoing preoperative evaluation for DRE
Interventions	Electrical cortical stimulation using 5 to 16 semirigid electrodes implanted per patient, in various cortical areas depending on the suspected origin of seizures
Outcome	Vestibular responses
Notes	Vestibular symptoms were induced on 44 anatomical sites in 28 patients. The patients experienced illusions of rotation (n = 30), translations (n = 6), or indefinable feelings of body motion (n = 8). Almost all vestibular sites were located in the cortex (41/44): in the temporal (n = 19), parietal (n = 14), frontal (n = 5), occipital (n = 2), and insular (n = 1) lobes. Among these sites, the authors identified a lateral cortical temporo-parietal area we called the temporo-peri-Sylvian vestibular cortex, from which vestibular symptoms, and above all rotatory sensations, were particularly easily elicited (24/41 cortical sites, 58.5%)
**Wiest (2004, Austria)** [29]	
Methods	Retrospective case report
Participants	1 patient undergoing preoperative evaluation for DRE due to an ependymoma
Interventions	Resective surgery
Outcome	Clinical manifestations, electrophysiologic findings, overall outcome
Notes	Electrical cortical stimulation of the precuneus could reproduce the vestibular sensations of linear self-motion and occasionally body tilts, which ceased after lesionectomy
**Isnard (2004, France)** [21]	
Methods	Retrospective case series
Participants	50 patients undergoing preoperative evaluation for DRE
Interventions	Direct electric stimulation of the insular cortex using implanted transopercular electrodes
Outcome	Responses that were evoked in the absence of any after-discharge
Notes	From a total of 125 responses, whole-body sensations (n = 5; 4%) occurred in six patients, as a sudden sensation of displacement of their body in space
**Mazzola (2006, France)** [17]	
Methods	Retrospective cohort study
Participants	14 patients undergoing preoperative evaluation for DRE
Interventions	Direct electrical cortical stimulation using transopercular electrodes exploring the following: the parietal opercular cortex (SII area); the suprasylvian parietal cortex (SI area); the insular cortex.
Outcome	Somatosensory and pain responses
Notes	Multiple non-somatosensory responses were obtained after stimulation of the insula, including vestibular sensations such as vertigo or horizontal rotation of the body
**Nguyen (2009, Canada)** [22]	
Methods	Retrospective case series
Participants	10 patients undergoing preoperative evaluation for DRE
Interventions	Direct current insular stimulation using depth electrodes, placed under direct vision after microsurgical opening of the Sylvian fissure
Outcome	Elicited responses
Notes	Electrical cortical stimulation performed in 9 of 10 patients with insular electrodes elicited, in decreasing order of frequency, somatosensory, viscero-sensory, motor, auditory, vestibular, and speech symptoms. Vertigo (3%) was rare
**Best (2010, Germany)** [30]	
Methods	Retrospective case report
Participants	1 patient undergoing preoperative stimulation for DRE
Interventions	Electrical cortical stimulation using three subdural electrodes to the temporal cortex and temporo-occipital junction, and one depth electrode toward the posterior insular cortex
Outcome	Clinical manifestations after cortical stimulation
Notes	Stimulation over the medial part of the STG elicited intracranial numbness, ipsilateral headache, and a subjective unidirectional vertigo
**Mazzola (2014, France)** [23]	
Methods	Retrospective case series
Participants	219 patients undergoing preoperative evaluation for DRE
Interventions	642 electrical stimulations of the insula, using stereotactically implanted depth electrodes
Outcome	Insular mapping of vestibular sensations
Notes	Vestibular sensations occurred in 7.6% of the 541 evoked sensations after electrical stimulations of the insula. They were mostly obtained after stimulation of the posterior insula, that is, in the granular insular cortex and the postcentral insular gyrus. The authors suggested a spatial segregation of the responses in the insula, with the rotatory and translational vestibular sensations being evoked at more posterior stimulation sites than other less definable vestibular sensations. No left–right differences were observed.
**Francione (2015, Italy)** [25]	
Methods	Retrospective case series
Participants	40 patients undergoing preoperative evaluation for DRE of surgical resections strictly confined to the parietal lobe
Interventions	Surgical resection of parietal lobe structures
Outcome	Anatomo-electro-clinical features and clinical outcome
Notes	The most frequent responses induced by electrical cortical stimulations in the parietal lobe structures, besides the simple somatosensory manifestations, consisted of vertiginous sensations and visual illusions. Vestibular sensations were clinically variable
**Mazzola (2017, France)** [24]	
Methods	Retrospective case series
Participants	222 patients undergoing preoperative evaluation for DRE
Interventions	669 electrical stimulations of the insula, using stereotactically implanted depth electrodes
Outcome	Clinical manifestations after insular stimulation
Notes	Somatosensory responses (61% of evoked sensations) including pain and visceral sensations (14.9%) were the most frequent, followed by auditory sensations (8%), vestibular illusions (7.5%), speech impairment (5%), gustatory, (2.7%), and olfactory (1%) sensations. The authors reported that although these responses showed some functional segregation (in particular, a privileged pain representation in the postero-superior quadrant), there was a clear spatial overlap between representations of the different modalities
**Yu (2018, China)** [26]	
Methods	Retrospective case series
Participants	43 patients undergoing preoperative evaluation for DRE
Interventions	Bipolar electrical stimulation with at least 1 electrode inserted into the insula or opercula via an oblique approach
Outcome	Clinical manifestations after electrical stimulation
Notes	A total of 6 responses involving vestibular symptoms out of 142 were related to left insula. Stimulation of the opercula evoked 10 vestibular responses; 2 cases with vertigo at 1.0 mA; 3 cases were sensations of deficiency of the contralateral limb or trunk at 1.0–2.0 mA; and 5 changes in the perception of the body’s location, at 1.0–4.0 mA
**Oane (2020, Romania)** [27]	
Methods	Retrospective case series
Participants	47 patients undergoing preoperative evaluation for DRE
Interventions	Depth electrodes in the cingulate cortex stereotactically placed using Leksel frame
Outcome	Cingulate cortex function and multi-modal connectivity mapping
Notes	Vestibular responses, defined as vertigo and dizziness, were elicited in 8 sites in the ACC, anterior MCC and PCC areas a24, p24, RSC, 31a, 23d. Connectivity analysis of these areas highlights important connections with lateral and mesial parietal regions, parietal operculum, and prefrontal cortex. Body perception responses, defined as altered perception related to location, gravity, or displacement of whole-body or body-parts, were elicited by stimulating 8 sites in MCC, PCC, and RSC.

(DRE, drug resistant epilepsy; ACC, anterior cingulate cortex; MCC, midcingulate cortex; PCC, posterior cingulate cortex; RSC retrosplenial cortex).

### 3.3. Elicited Vestibular Responses (Q1)

The vestibular responses elicited after electrical cortical stimulation can be grouped into four categories: vertigo and/or dizziness without ocular movement [22,24,25,29], perception of body motion [24], vertigo and/or dizziness with ocular movement, and other indefinable and/or ill-defined responses. Vertigo has been described by the patients as a sensation of “dizziness allover” [2], “sensation of full head” [2], “things turning around” [2], and “rocking and alternating tilting sensations of the body and the environment” [29]. Other patients described vertigo as “head-spinning” [12], “feeling of drunkenness” [12], “unsteadiness” [12], and “disequilibrium” [25]. Body movement, either as rotation or in the form of displacement, has been described by the patients as a feeling of “sinking” [2], “moving around” [2], “floating” [18,25], “body swaying” [20], “rolling off the table” [20], “sliding toward the lower end of the bed” [28], and “oscillations of the body” [12]. Patients may even feel of “being pushed”, “being attracted to the right side” [12], or “falling flat” [25]. A subgroup of these patients complained of disturbed body image, including “a twisting sensation of the contralateral extremity” [20] or “alternating tilting sensation of the patient’s own body” [29]. The third group of manifestations includes dizziness associated with conjugate eye deviation to the opposite side [2]. Finally, among the indefinable or ill-defined perceptions are “sensation of full head” [2], “vague cephalic sensations” [19], “pulsations inside the head” [25], “lightness”, and “levitation” [12,32]. It has to be mentioned that, in the vast majority of the cases, the elicited semiology after stimulation was concordant to the symptoms occurring during the patient’s aura.

### 3.4. Anatomic Correlations (Q2)

Regarding the anatomic location of the vestibular cortex, the parietal lobe is the most studied target in seven studies. Stimulation of the parietal lobe with the somatosensory cortex elicited vestibular responses in up to 11% of the cases [20]. These responses were perceived by the patient as a sense of rotation or bodily displacement [2]. Richer et al. reported that stimulation of rolandic and parietal brain regions induced sensations of body floating, that was not accompanied by actual movement [18]. Salanova et al. postulated that among 80 patients who underwent intraoperative electrical cortical stimulation of the parietal lobe, one patient reported a sensation of rolling off the table; two patients developed a disturbance of body image; one patient described a twisting sensation of the contralateral extremity; another one reported a swinging sensation after stimulation of the non-dominant inferior parietal lobule [20]. Blanke et al. focused on the electrical cortical stimulation of the left inferior parietal lobule [28]. Vestibular responses were evoked at two sites, posterior to the somatosensory representation of the lower face and tongue, and superior to the posterior language cortex. The Talairach coordinates of these two areas corresponded to the left anterior part of Brodmann’s area 40 (anterior part of the inferior parietal lobule) [28]. These sensations were described as if the patient was “sliding towards the lower end of the bed” [28]. After stimulating at a more caudal location, the patient described a sensation of alternating side to side rotation of his entire body [28]. In another study by Wiest et al., the electrical cortical stimulation of the precuneus induced sensations of “rocking”, “swaying”, or like “being on a boat”, accompanied by the feeling of being pushed or pressed to the left side [29]. According to Francione et al., the responses most frequently induced by electrical cortical stimulation of the parietal lobe, excluding the simple somatosensory manifestations, consisted of vertiginous sensations and visual illusions [25]. The majority of the vestibular symptoms were subjective vertigo, and the remaining varied from head and body oscillations to the sensation of falling flat and disequilibrium [25]. The authors reported no specific topographic pattern in regard to the patient’s semiology [25]. The insula constituted the second most studied cerebral structure in six studies. Vertigo occurred after insular stimulation with a reported frequency ranging from 3% to 8% [22,24]. Isnard et al. studied the clinical responses after 144 insular stimulations with stereotactically implanted depth electrodes in 50 consecutive patients with temporal lobe epilepsy [21]. A total of 139 evoked clinical responses from 125 stimulation sites were collected. Whole-body sensations represented 4% of the evoked responses, with six patients reporting a sudden sensation of displacement of their body in space, such as a brisk forward projection, a vertical or horizontal rotation of their body, or a sensation of levitation [21]. Mazzola et al. provided a functional mapping of vestibular responses after electrical stimulation of the human insular cortex [23]. The authors reported that vestibular sensations occurred in 7.6% of their cases, particularly after stimulation of the granular insular cortex and the postcentral insular gyrus [23]. Rotatory and translational illusions occurred at more posterior sites than other less definable symptoms, suggesting spatial segregation [23]. Equally important, the authors postulated no left–right differences in their cohort related to the elicited symptoms and the location of stimulation [23]. The remaining elicited somatosensory sensations included pain (14.9%), auditory sensations (8%), speech impairment (5%), gustatory (2.7%), and olfactory sensations (1%) [24]. In another study, Mazzola et al. compared the responses elicited after the stimulation of the parietal opercular cortex (SII area) with those from the suprasylvian parietal cortex (SI area) and the insula [17]. Multiple types of non-somatosensory responses were obtained from the insula, including vestibular sensations such as vertigo or horizontal rotation of the body [17]. Yu et al. studied the functional anatomy of the insular lobe and the opercula in 43 patients with drug-resistant epilepsy using direct current electrical stimulation [26]. All were related to the left insula [26]. Dizziness was evoked by the stimulation of the superior portion of the posterior long gyrus [26]. The remaining two patients experienced limb deficiency after stimulation of the anterior long insular gyrus [26]. Another potential location of the vestibular cortex is the temporal lobe and the temporal peri-Sylvian cortex, according to three studies. Penfield reported that stimulation of the cortex adjacent to the transverse gyrus of Heschl produced sensations of dizziness, described as ‘sinking”, “swinging”, and “moving all around” [2]. In another study, Fish et al. recorded the clinical responses to electrical stimulation of the temporal and frontal lobes in 75 patients with drug-resistant epilepsy [19]. Vague cephalic sensations or dizziness occurred in six patients after stimulation of the amygdala and hippocampus without afterdischarges [19]. Of note, the authors clarified that these sensations were not true vertigo [19]. Kahane et al. assessed the cortical areas with vestibular inputs via electrical stimulation in 260 patients with partial epilepsy [12]. Stimulation of 44 anatomical sites in 28 patients produced vestibular symptoms [12]. The temporal lobe was involved in 19 cases, the parietal lobe in 14, the frontal lobe in 5, the occipital lobe in 2 cases, and the insula in 1 [12]. The reproduced vestibular symptoms were rotatory illusions in 30 patients, translation in 6 cases, and undefined body motion in 8 patients [12]. The authors identified a lateral cortical temporo-parietal area, the temporal peri-Sylvian cortex, as the vestibular cortex from which vestibular symptoms, and mainly rotatory sensations, were particularly easily elicited [12].

Electrical stimulation of the occipital lobe or at its junction with the temporal lobe has been implicated in eliciting vestibular responses. Penfield reported that stimulation anterior to, or even in, the occipital cortex might cause dizziness accompanied by a conjugate deviation of the eyes to the opposite side [2]. In a case-report, Best et al. noted that electrical stimulation of the posterior part of the middle temporal gyrus at the temporo-occipital junction induced strictly horizontal nystagmus, with its slow phase toward the stimulated hemisphere [30].

Limited data demonstrated that the frontal supplementary somatosensory and the posterior cingulate areas may represent parts of a more extensive vestibular cortical network and may also be implicated in the pathogenesis of vertiginous epilepsy [27]. Therefore, stimulation of any of the involved areas may well produce vertigo sensation [27]. Oane et al. postulated that vestibular responses were elicited by high-frequency stimulation of the cingulate cortex and concluded that the posterior cingulate gyrus seemed to be connected with the visual areas, mesial and lateral parietal, and temporal cortex [27]. A schematic representation of all identified stimulation sites is provided in Figure 2 and Figure 3.

### 3.5. Left vs. Right-Sided Stimulation

Kahane et al. reported that stimulation areas from which vestibular responses were elicited were widely distributed on the right and the left hemispheres [12]. Equally important, no significant differences in the type of evoked sensations were noted between right- or left-sided insular stimulations [24]. The side of the falling sensation was not related to the side of insular stimulation [24]. Similarly, the direction of the translation did not seem to depend on the side of the stimulation [23]. However, stimulations of either the right or the left insula evoked sensations of rotation in a clockwise direction [23]. Thus, according to Penfield’s tentative conclusion, the pathway of vestibular sensory information makes a detour from the thalamus out to the cortex, where the vestibular area is next to the auditory area in the superior temporal convolution of both hemispheres [2].

### 3.6. Intensity and Frequency of Stimulation (Q3)

There was evidence that the vestibular responses became more robust with increasing stimulation current intensity. Blanke et al. reported that stimulation of the inferior parietal lobule at low intensities (4.5–5.5 mA, 2 s duration) resulted in a sensation of sliding towards the lower end of the bed [28]. With a stepwise increase in the current intensity (5.5–8.0 mA, 2 s), there was a sensation of rolling to the right with an urge to hold on to something to prevent from falling out of the bed [28]. Eventually, the patient felt that objects in the environment turned with him to the right (10.5 mA, 5 s), accompanied by an unpleasant sensation [28]. In another study, stimulation over the medial part of the superior temporal gyrus at low intensities (8–12 mA) induced blurring of the vision [30]. With an intensity of 14 mA, the patient experienced an intracranial numbness, ipsilateral headache, and subjective unidirectional vertigo [30]. Of note, Kahane et al. noted that both low- (1 Hz) and high-frequency (50 Hz) stimulation induced vestibular responses [12]. However, the former proved to be more effective, particularly in the temporal, occipital, and insular areas [12].

### 3.7. Other Non-Vestibular Manifestations (Q4)

Seven non-vestibular manifestations may occur during the stimulation of vestibular associated cortex. No study reported actual movement of the body or the limbs. A few patients reported eye jerk, blurred vision, diplopia, or oscillopsia without deviation or nystagmus, causing difficulty in focusing on an object [2,12,22,25]. Furthermore, somatosensory responses, including non-specific pain, were commonly recorded, particularly with stimulation of the parietal lobe and the insula [17,21,22]. Viscero-sensory and autonomic manifestations, including nausea and vomiting, pharyngo-laryngeal constriction, tachycardia, abdominal buzz, a sensation of a rising warmth in the digestive system, or gustatory illusions, were also reported in some cases [12,17,22,26]. Frequently, the vestibular responses were accompanied by an unpleasant emotional sensation, fear, and sensations of unreality [17,29]. Objectively, a slowing during counting and naming tasks without specific speech problems was occasionally noted [17,26,28].

## 4. Discussion

The current systematic review focused on the vestibular responses after electrical cortical stimulation. Throughout the years, vestibular sensation secondary to electrical stimulation of the cortex has been generally overlooked, although a few reports have mentioned its existence [6,7,8,11,28,30,32,33,34]. Penfield and Jasper firstly described this phenomenon and introduced the term “labyrinthine sensation” in patients with medically refractory epilepsy after central and temporal areas direct, intraoperative, electrical stimulation [2]. In our study, two main types of vestibular responses were identified, including vertigo and perception of body movement. The latter could be either rotatory or translational. Electrical cortical stimulation of the temporal structures elicited mainly vertigo, while stimulation of the parietal lobe was associated with perceptions of body movement. Stimulation of the occipital lobe produced vertigo with visual manifestations. There was evidence that the elicited vestibular responses were aggravated with the increase in the stimulation current intensity. Low-frequency proved to be more effective than high-frequency stimulation in eliciting vestibular responses, particularly in the temporal, occipital, and insular areas. Seven non-vestibular responses could be elicited after stimulation of the so-called vestibular cortex, including somatosensory, viscero-sensory, and emotional manifestations.

Several experimental electrophysiological studies have been performed for identifying and outlining vestibular cortical areas in various animal models [6,7,8,9,35,36,37,38,39,40]. A series of studies in primates have shown that vestibular information is processed through certain cortical areas: (a) the parieto-insular cortical junction (PIVC), located at the posterior end of the insula, (b) Brodmann’s area 2v, located at the tip of the intra-parietal sulcus, (c) Brodmann’s area 3aV, located in the central sulcus, (d) Brodmann’s area 7, located in the inferior parietal lobule, and (e) Brodmann’s area 6 [7,8,10,41]. It has been demonstrated that all parts of the vestibular cortical circuit are interconnected, and that the PIVC represents the core of this vestibular cortical network [12,21,37,42]. Similarly, a vestibular cortical circuit including the same anatomical areas has been described for cats [5]. Several animal electrophysiological studies have shown that all parts of the vestibular cortical circuit receive bilateral input from the vestibular nuclei, while they send direct projection fibers back to these nuclei [8,10,11,35,43]. There is a growing body of evidence (microelectrode stimulation and recording studies) demonstrating that these vestibular cortical areas are clusters of multisensory neurons [28,32,41,44]. Indeed, the microscopic cyto-architecture of these cortical areas is characteristic of multisensory rather than unimodal sensory cortex [9,44].

The existence of a similar vestibular cortical network in humans can be extrapolated from the available experimental animal data. However, human data are far less documented and very limited. The existing human data have either disruption sources, as in patients with structural lesions of certain anatomical areas, or activation source from imaging and electrophysiological studies mainly from epileptic patients [1,10,21,28,30,32,36]. However, newer imaging modalities such as functional MRI (fMRI), Positron Emission Tomography (PET), single photon emission computed tomography (SPECT), and near infra-red spectroscopy (NIRS) have been recently employed in order to identify and delineate any vestibular cortical areas [44,45,46,47,48,49,50,51,52,53].

There is a significant body of fMRI data, which is usually combined with either galvanic or caloric stimulation [45,46,48,50,51,53,54]. Galvanic stimulation particularly, in combination with fMRI, may provide valuable information in physiological and pathological conditions regarding the structure of the vestibular network [55]. Areas such as PIVC, posterior insular cortex (PIC), middle temporal gyrus (MTG), cingulate sulcus (CS) and gyrus, peri-Sylvian cortex, and supramarginal gyrus are activated in fMRI, indicating that they play a significant role in the processing and integration of vestibular stimuli. This concept is also supported by the fact that these areas provoked either vertigo symptoms or perception of body movement during electrical cortical stimulation as shown in our current study [45,46,48,50,51,53,54]. Aedo-Jury et al. [48] also suggested that V6 cortical visual area is also activated during galvanic stimulation. In addition to these areas, Beer et al. [54] suggest that precuneus motion (PcM) and regions adjacent to the callosum also respond to vestibular cues.

In regard to the emerging PET data, this is scarcer. Brandt et al. [46] and Devantier et al. [49] used PET along with galvanic/caloric stimulation and rotation or sideways movement, respectively. In the first case, PIVC was activated, while in the second one, the authors postulated that the Heschl’s gyrus is part of the vestibular cortex.

Another imaging modality recently employed for localizing vestibular cortex is NIRS [47,52,56]. Temporo-parietal junction (TPJ), MTG, and supramarginal gyrus are all areas identified via both NIRS and electrical cortical stimulation as parts of the vestibular cortex. Nguyen et al. [52] also added intraparietal sulcus to these areas.

Few other studies have attempted to localize the vestibular cortex using other various advanced MR-based imaging techniques [57,58,59,60,61,62]. Indovina et al. [63] used the Human connectome project along with tractography methods to support the idea that the posterior peri-Sylvian cortex and PIC are implicated in the vertiginous network. These two areas, along with CS, MTG, and somatosensory cortex, were also indicated by Li et al. [64], who combined cases of brain lesion with vestibular symptoms and electrical cortical stimulation. Patients with vestibular migraine could also provide valuable information regarding the outlining of the vestibular network [65,66,67,68]. Two of these studies [67,68] used MRI volumetric studies and calculated the gray matter volume in patients with vestibular migraine. The volume of the prefrontal cortex, middle frontal gyrus, posterior insula-operculum regions, inferior parietal gyrus, supramarginal gyrus, PIVC, and precuneus was decreased. Interestingly, Raiser et al. [69], in their study, suggest that the cortical vestibular network has a right-sided dominance. There is plenty of evidence in the pertinent literature demonstrating that there is lateralization of the vestibular cortex. It is usually located in the dominant hemisphere, matched with the speech dominance [70,71].

There is a growing body of evidence, from the employment of advanced imaging and hybrid imaging-electrophysiological studies, indicating that there is a quite extensive and complex vestibular network, rather than few, discrete cortical vestibular areas. Anatomical areas such as the PIVC, PIC, MTG, Heschl’s gyrus, peri-Sylvian cortex, inferior parietal lobule, and supramarginal lobule are persistently implicated in the processing of vestibular stimuli. Thus, this emerging data indicate that these areas play a significant role in vestibular processing and integration. Undoubtedly, a more accurate outlining and better understanding of the vestibular network would help us develop more efficacious treatment strategies for vestibular pathologies.

It must be emphasized that our current review has some important limitations. Initially, it is based on a limited number of low and very-low quality of evidence articles. We have included studies of variable study design, including case-series. Moreover, vestibular manifestations are subjective responses, which cannot be verified and quantified; thus, their prevalence cannot be accurately estimated. Also, various intraoperative triggering events could elicit vestibular responses, resulting in false-positive recordings. Furthermore, there were no quantitative data to perform a meta-analysis. The current review is characterized by all the potential limitations of literature reviews, including error propagation. It has also to be mentioned that we have not considered, in our review, any possible inter-study heterogeneity.

## 5. Conclusions

Evidence from electrical cortical stimulation studies indicates that the vestibular cortex could be localized in the temporal, parietal, and the temporo-occipital junction. It seems that the vestibular cortex has the form of an extensive network rather than a discrete, limited, and well-demarcated cortical location. The elicited responses range from vertigo to body movement perceptions. On its own, electrical stimulation might not be enough to draw conclusions as robust and definitive as those drawn in conjunction with other advanced imaging and functional modalities. Further high quality and accuracy electrical stimulation studies are required for identifying and precisely outlining all the possible underlying subcortical connections.

## Figures and Tables

**Figure 1 brainsci-14-00075-f001:**
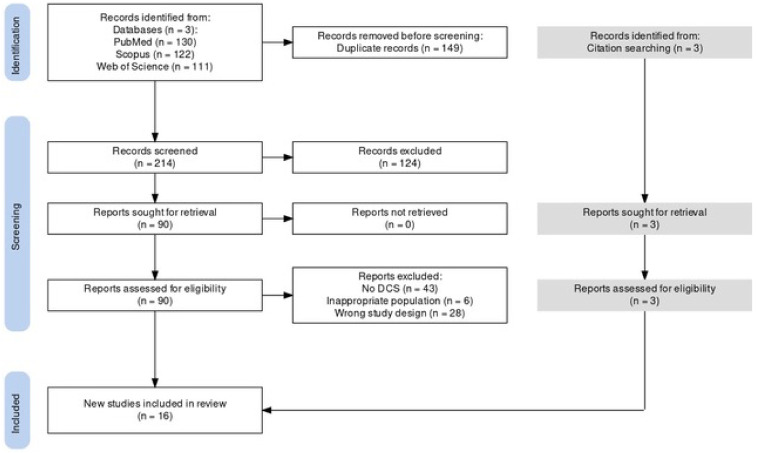
Schematic flowchart of our literature search.

**Figure 2 brainsci-14-00075-f002:**
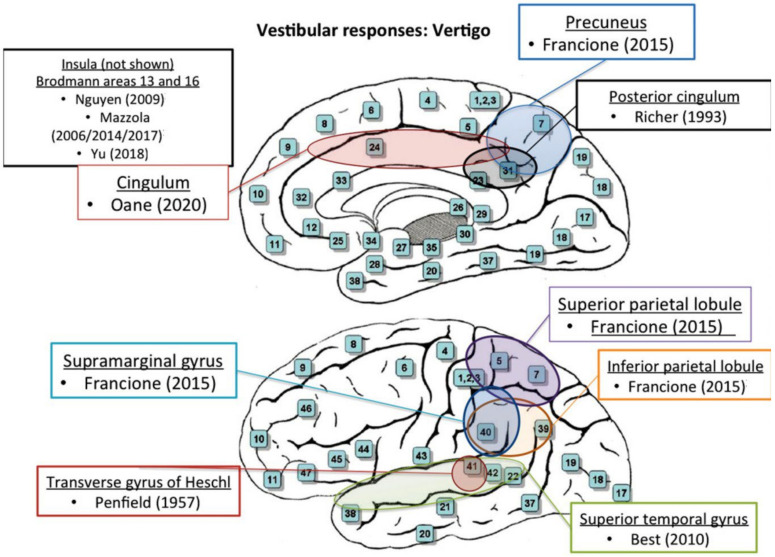
Vertigo. Schematic topographical depiction of all stimulation sites eliciting vertigo after stimulation [2,17,18,22,23,24,25,26,27,30].

**Figure 3 brainsci-14-00075-f003:**
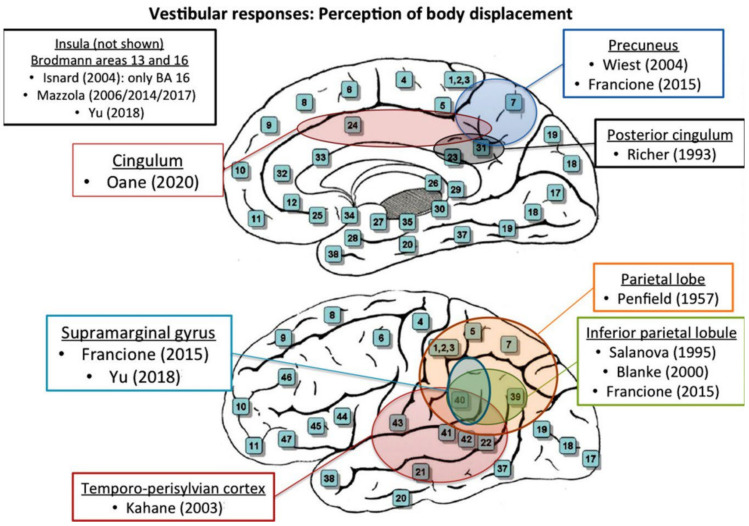
Perception of body movement. Schematic topographical depiction of all stimulation sites eliciting body displacement responses [2,12,17,18,20,21,23,24,25,26,27,28,29].

**Table 1 brainsci-14-00075-t001:** Search strategy according to the PICOT protocol.

Frame	P (Participants)	I (Intervention)	C (Comparator)	O (Outcome)	Time
**Mesh terms**	Any	Electrical stimulation of brain structures	None	Vestibular response	Intra-operatively or during the preoperative evaluation for drug resistance epilepsy (DRE)
**Search**	**PubMed:** (((direct current stimulation) OR (electrodes)) AND ((Vestibular) OR (vertiginous))) AND (cortex) **Scopus:** ((TITLE-ABS-KEY (vestibular) OR TITLE-ABS-KEY (vertiginous))) AND ((TITLE-ABS-KEY (direct AND current AND stimulation) OR TITLE-ABS-KEY (electrodes))) AND (TITLE-ABS-KEY (cortex)) **Web of Science:** https://www.webofscience.com/wos/woscc/summary/accf5915-f542-43db-ac83-d52ad90ff489-be1316c5/relevance/1 accessed on 30 November 2023				
**Exclusion Criteria**	Irrelevant title or abstract, irrelevant full-text, editorial, reviews, meta-analysis, neonatal studies, experimental/non-human studies, non-English studies, stimulation other than electrical (caloric, galvanic), responses not including vertigo or perception of body movement				
**Sources**	Databases (PubMed, Scopus, Web of Science) Reference list				
**Time limits**	The search period: any until July 2023			Last search: 30 November 2023	

## Data Availability

No new data were created or analyzed in this study. Data sharing is not applicable to this article.

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
