# Peer review of "Localization of Vestibular Cortex Using Electrical Cortical Stimulation: A Systematic Literature Review"

_brainsci, 2024, doi:10.3390/brainsci14010075_

Round 1
Reviewer 1 Report
Comments and Suggestions for Authors
The article is interesting, but with many limitations that have already been addressed by the authors in the appropriate section. Few corrections should be done:
The abbreviation “DRE” in table 1 should be introduced in full (drug resistant epilepsy (DRE) the first time it appeared in text, or keynote below table 1 like in table 2.
Page 3 line 107 : spelling mistake : “identified though citation”, Correction : “identified through citation”
In figure 2 & 3 some references had year between brackets e.g. Yu, (2018), others had year as well as numerical reference after brackets e.g. Wiest, (2004)29. Only one way of writing references should be used.
Reviewer 2 Report
Comments and Suggestions for Authors
A very nice systematic review on the localisation of the vestibular cortex. I accept and fully agree with the limitations of the study as the authors have mentioned. This topic will be of relevant interest and of high importance to future work on understanding this diffuse vestibular network better and will help with targeting treatment plans for vestibular disorders including PPPD, where often there are a lot of non-specific, non-pathognomonic description of dizziness. When we can better understand the anatomic correlates, we can then formulate treatment plans targeting these areas better.
Just one minor correction
Line 320: SPECT. Good to spell it out in full as Single-Photon Emission Computed Tomography (SPECT) to be consistent with the rest of the imaging modalities listed.
Congratulations.
Reviewer 3 Report
Comments and Suggestions for Authors
The authors conducted a systematic literature review to summarize the existing knowledge on vestibular cortical localization through electrical cortical stimulation, identify any potential anatomic correlations, determine the potential role of the employed stimulation intensity and frequency in eliciting certain vestibular responses, and identify the common vestibular manifestations following such stimulation. After systematically searching, filtering, and qualifying, the authors selected and included 16 studies that reported on the electrical cortical stimulation of cortical areas for the review article. The authors concluded that the vestibular cortex could be localized in the temporal, parietal, and temporo-occipital junctions, forming an extensive network rather than a discrete, limited, and well-demarcated cortical location. The elicited vestibular responses range from vertigo to body movement perceptions. The manuscript is well-structured and well-written. I have some comments on this manuscript.
The localization of the vestibular cortex has been a subject of investigation for an extended period and has recently achieved significant milestones, thanks to the development of modern functional imaging and brain activity monitoring techniques. Given the conventional understanding that the vestibular cortex responds to stimuli in the vestibular organs, localization of the vestibular cortex becomes more accurate when utilizing stimuli at the vestibular end-organs. Among these stimulating methods, galvanic vestibular stimulation (GVS) is simple, common, non-invasive, and useful to apply. GVS activates both primary otolithic and semicircular canal neurons and their cortical projections to vestibular cortex [Utz et al. (2010) Electrified minds: Transcranial direct current stimulation (tDCS) and galvanic vestibular stimulation (GVS) as methods of non-invasive brain stimulation in neuropsychology—A review of current data and future implications. Neuropsychologia; Dlugaiczyk et al. (2019) Galvanic vestibular stimulation: From basic concepts to clinical applications. J. Neurophysiol]. The present review manuscript is dedicated to exploring the localization of the vestibular cortex through electrical cortical stimulation. Could you please provide a more detailed description of the similarities and differences in localizing the vestibular cortex when stimulated by either “electrical cortical stimulation” or “galvanic/caloric vestibular stimulation”? Additionally, what is the primary significance of the “vestibular cortex” localized through electrical cortical stimulation?
Is there any difference between "electrical cortical stimulation" and "cortical electrical stimulation"? Both terms appear in this manuscript; could you standardize the usage of the term?
Considering that vertigo is a specific type of dizziness and can be challenging to accurately discern in a systematic literature review, do you feel confident using the term 'vertigo' instead of 'dizziness'?
Line 5: Please remove “and” in the phrase “MD and PhD”
Line 35: The sentence “The importance of the vestibular system in various human body functions cannot be overemphasized" is confusing. Please rephrase this sentence.
Line 144: It seems there are duplicate words in the phrase: “levitation” [12]. levitation”[32]?
Line 352-353: Regarding the sentence "Interestingly, Raiser et al., in their study, suggest that the cortical vestibular network has a right-sided dominance". Actually, vestibular lateralization has indeed been distinctly elucidated in various functional neuroimaging studies conducted on both animals and humans. Specifically, vestibular dominance aligns with the same side as handedness in humans, with right-sided dominance in right-handers and left-sided dominance in left-handers [Dieterich et al (2003) Dominance for vestibular cortical function in the non-dominant hemisphere. Cereb Cortex; Dieterich and Brandt (2018). Global orientation in space and the lateralization of brain functions. Curr Opin Neurol.].
